# Exploring the psycho-social burden of infertility: Perspectives of infertile couples in Cape Coast, Ghana

**Abdoul Azize Diallo** [1,2]*, **Prince Justin Anku**[2], **Rhodalyn Adwoa Darkoa Oduro**[1]

**1** Department of Obstetrics and Gynaecology, School of Medical Sciences, College of Health and Allied Sciences, University of Cape Coast, Cape Coast, Ghana, **2** Department of Population and Health, Faculty of Social Sciences, College of Humanities and Legal Studies, University of Cape Coast, Cape Coast, Ghana

* diallo.azize@uccsms.edu.gh

**Data Availability Statement:** All relevant data are within the manuscript. However, the raw interview transcripts cannot be made publicly available due to ethical concerns and the reassurance of confidentiality given to the couples before they consented to the study. Excerpts of the transcripts

## Abstract

Infertility is a complex and often sensitive issue with far-reaching psycho-social ramifications for couples and their families. This study therefore seeks to delve into the psycho-social burden of infertility in Cape Coast, a major city in Ghana. Specifically, we explored the impact of infertility on the psychological and social health of infertile couples receiving fertility treatment. It also delves into the strategies they adopt to cope with their conditions. The study employs a qualitative approach to inquiry using phenomenology as a study design to explore the experiences of the study participants. In-depth interviews were conducted using interview guides, voice recorded and transcribed verbatim. Both inductive and deductive/framework coding techniques were used to code the data leading to the generation of themes and sub-themes. The results show that most of the study participants dealing with infertility faced psychological burdens from different sources including their families, society and themselves. These burdens take a toll on their mental health, pushing them into a state of desperation and depression. It was, however, revealed that infertile couples are able to cope with the help of their family, spouses and the church. Curiously, some of them opt for withdrawal from social events as a coping mechanism. Infertility exerts an enormous negative psycho-social impact on affected couples, especially women. The family and society serve as the main sources of stressors for infertile couples. Therefore, programmes that are aimed at fertility treatment should deliberately consider addressing the psychosocial burden of infertility through education targeting actors, especially interpersonal-level actors.

## Introduction

Infertility is an important reproductive health issue that affects millions of individuals and couples globally [1]. It is the inability of a sexually active, non-contracepting couple to achieve pregnancy after 12 months of regular, unprotected penial-vaginal intercourse. Globally, it is estimated that about 12 per cent of couples in their reproductive age suffer from infertility [2]. The impact is on the physical, emotional and social health of the couple, with women largely bearing the brunt of society [3].

can be made available upon request through the Department of Obstetrics and Gynaecology, School of Medical Sciences, University of Cape Coast. Email: info@uccsms.edu.gh.

**Funding:** The author(s) received no specific funding for this work.

**Competing interests:** The authors have declared that no competing interests exist.

According to Nukunya [4], the traditional Ghanaian society is pro-natal. This implies that the ultimate purpose of marriage is to produce children who will continue the family name. Hence, it is quite rare to see a couple in Ghana choose voluntary childlessness. As such, it is postulated that married couples without children, most likely are dealing with issues of infertility [5]. The prevalence of infertility in a Ghanaian setting has been estimated to be 12.3 per cent with secondary infertility being higher [6].

Infertility can have devastating consequences, with individuals and couples often experiencing stigma, shame, anxiety, depression, ostracism, and low self-esteem [7, 8]. In cultures where the continuation of the family name through the birth of a child can safeguard marriage, guarantee inheritance rights, and serve as social security in old age, the stakes could not be higher [9].

Infertility is not just a medical condition; it is a deep-seated personal and socially embedded phenomenon. Whereas advancements in reproductive medicine have expanded treatment options [10, 11], the psychological and emotional dimensions of infertility are yet to be fully understood. Fertility treatments can be financially burdensome, physically demanding, and emotionally draining on couples [12]. Fertility treatment in Ghana can be financially burdensome as it is not covered under the National Health Insurance Scheme (NHIS) and couples have to fully pay for all services and treatment [13]. By unravelling the psychosocial contexts of infertility, this paper aims to uncover the nuances, offering insights that may guide clinicians, counsellors, and support groups in providing more comprehensive care. Our study is framed around the Social Ecological Model (SEM).

## Theoretical framework: Social ecological model

The Social Ecological Model (SEM) was originally developed by Bronfenbrenner in the 1970s and has since been adapted in several forms to study complex social issues. The SEM takes into account the individual (infertile individual or couples), and their relationship/interactions with people, organizations, and their community at large [14]. There are five levels to the model–Individual, Interpersonal, Organizational, Community, and Public Policy [15]. The individual level is concerned with an infertile individual/couple's knowledge and skills about their condition. Knowledge about infertility helps individuals/couples to better understand their condition and how to deal with it. However, knowledge alone is not enough to help individuals/couples deal with their infertility. But it helps a lot in influencing key attitudes and decisions individuals and couples make concerning their infertility.

The interpersonal level has to do with infertile couple's relationships with other people–family, friends, colleagues, health providers and so on. Here, the parents, close relatives, or service providers can have regular talks with the infertile individual/couple about treatment options and procedures in an attempt to find solutions. Making regular counselling services available for the couple might help a great deal. At the organisational level, the issues are concerned with rules, regulations, and informal structures that may constrain or promote the efforts of infertile couples in addressing their fertility challenges. According to SEM, a community refers to the health facilities and the various organizations (both governmental and nongovernmental) in the area of fertility and reproductive health. These facilities/organizations can pool resources and ideas together to address the problem of infertility and help couples live a more fulfilling life. At the final level is the public policy/regulations regarding fertility treatment. It consists of the wider environment that draws heavily on policies, attitudes, ideologies, culture, and beliefs that have indirect consequences on the other levels of the social system and the infertile couples. This level of the SEM is important because it affects the other levels of the social system. Policies, attitudes, and cultures that make alternative options and

solutions to infertility (e.g., In vitro fertilization [IVF], Intrauterine insemination [IUI], and Egg and Sperm donation among others) more accessible and acceptable are critical.

## Materials and methods

This is a qualitative study that employed phenomenology as a study design to explore the social and psychological dimensions of couples dealing with infertility. Specifically, the study made use of descriptive phenomenology to describe the essence of infertility, capturing the commonalities of the lived experiences. In line with this design, in-depth descriptions of participants' experiences were collected through interviews. This design was deemed appropriate since the focus of the study is to unravel the psycho-social burden of the phenomenon based on the lived experiences of infertile couples.

This study was conducted in Cape Coast, the regional capital of the Central region of Ghana. Cape Coast is a cosmopolitan city whose population is diverse due to its positive net migration [16]. About half of the participants interviewed in this study are residents of Cape Coast, and the remaining were referrals from other districts of the central region which is mostly populated by the Fante people, who are predominantly Christians [16].

The study population are all infertile couples who attended a fertility clinic in a tertiary facility in the study region during the study period. However, we excluded couples with infertility of less than a year as well as couples with secondary infertility. These people were excluded due to the focus of the study and the operational definition of infertility since couples who are unable to achieve pregnancy under a year are not considered infertile. Also, those with secondary infertility may have a different psycho-social burden as compared to women with primary infertility.

A purposive sampling technique was used to select the study participants. Data collection and preliminary analyses were done simultaneously to determine data saturation. The data collection and analyses were iterative and progressive so that saturation could be properly assessed. Saturation was reached at the 8th interview. However, two additional interviews were conducted to affirm meaning saturation [17]. An interview guide was used for the data collection. All interviews were recorded on audiotape and then transcribed verbatim into Microsoft Word files. The few interviews that were in the local language were first translated into English by a language expert and played back to ensure that meanings were not lost during the translation process.

A total of ten (10) infertile couples were purposefully recruited for the study. Data was collected between 1st February 2023 and 28th February 2023 during clinic days. Participants were interviewed together as a couple unit. At the clinic, potential participants (couples) were approached, the purpose of the research was explained to them and their consents were sought. Couples who consented to participate in the study were then taken to a private room for the interview. A highly flexible semi-structured interview guide was used to elicit the responses from the participants. A relaxed conversation-like atmosphere was created and participants were encouraged to feel free to talk about the issues as much as possible. The duration of the interviews varied between 30 minutes to 60 minutes. All interview sessions were voice recorded and transcribed after the interview. In addition, the gestures and emotions exhibited by the participants were noted and incorporated into the transcripts. This allows us to better appreciate the contexts to adequately capture the psychological realities of the study participants.

The audio-recorded data was translated into English, in cases when the couples expressed themselves in the native language, i.e. (Asante Twi), and then transcribed verbatim to capture the exact responses of the study participants. We followed Morse, [17] strategy for qualitative

data analysis. This included reading data thoroughly (more than once). This was then followed by inductive coding to identify areas of interest and clustering them by similar ideas. Then themes and categories were generated. The key tenets of the SEM which was used in the development of the theoretical framework, guided the second stage of the data analyses. The themes and categories that fell under the key tenets of the SEM were adequately sorted. Quotes were then attached to the various themes and categories.

To ensure reliability in the codes, we employed an inter-rater coding mechanism where PJA and AAD independently coded the data. All authors then met and reviewed the codes and the few inconsistencies in the codes were resolved together. The decision not to initially employ deductive coding (based on a predetermined coding scheme) was necessary to avoid imposing the theoretical framework on the data but to allow themes to emerge freely. The results are presented based on the emerged themes and categories and discussed within the ambit of the Social Ecological Model (SEM) which was used as a theoretical framework for the study.

This research has received ethical approval from the Cape Coast Teaching Hospital Institutional Review Board (REF: CCTHERC/EC/2023/011). Written informed consent was obtained from all the study participants before the interviews. The transcripts were anonymized by bracketing all names and identities. Unique codes were then assigned to each participant's transcript for easy identification and retrieval of information.

## Results

In total, 10 infertile couples were interviewed. Table 1 summarizes the background characteristics of the couples.

From the data analysis, three themes and nine sub-themes emerged. Table 2 summarizes the themes and subthemes (code book).

### Sources of psychological pressures

**Stressors from family members.** The family was found to be a common source of stressors for infertile couples. Family members either knowingly or unknowingly through their actions and inactions put infertile couples under enormous stress. Some infertile couples recounted various acts of pity exhibited towards them by their families. Often, family members either directly or indirectly show pity by trying to suggest solutions to the couple's predicament. This is usually appreciated by the couples in the early stages of their infertility. However,

**Table 1. Characteristics of the participants.**

| Participating couples Code | Age (male) | (female) | Occupation (male) | (female|) | Years of infertility |
|---|---|---|---|---|---|
| 001 | 41 | 35 | Driver | Trader | 8 |
| 002 | 42 | 37 | Carpenter | Trader | 9 |
| 003 | 37 | 29 | Teacher | Lab technician | 3 |
| 004 | 35 | 27 | Lecturer | Nurse | 4 |
| 005 | 35 | 29 | Lecturer | Banker | 5 |
| 006 | 38 | 35 | Trader | Trader | 7 |
| 007 | 34 | 30 | Lecturer | Student | 4 |
| 008 | 40 | 35 | Mason | trader | 5 |
| 009 | 38 | 32 | Trader | hairdresser | 9 |
| 010 | 40 | 33 | Teacher | Teacher | 5 |

Fieldwork, 2023

Table 2. Themes and subthemes from the infertile couples.

| Number | Themes | Sub-themes |
|---|---|---|
| 1 | Psychological stressors/pressures' source | 1. Family members<br>2. Self-inflicted<br>3. Society/Community<br>4. The Church |
| 2 | Psychological impacts of infertility | 1. Desperation<br>2. Depression |
| 3 | Coping strategies | 1. Active avoidance of people or social events<br>2. Family support<br>3. Self-motivation<br>4. Church organizations/activities |

as their search for a solution drags on after many attempts, couples become frustrated and stressed out when family members continue to offer possible solutions. One of the participants recounted:

*. . . For instance, you go to a funeral and someone walks up to you with herbs to solve infertility problems. To them, they think they are helping you but they are rather compounding the problem. It gives us so much stress when people randomly come to us with purported solutions.* (Couple_003; 3 years of infertility)

Whereas most family members unknowingly cause infertile couples stress, some family members sometimes do it deliberately through their actions and inactions. The narratives from some participants revealed that some close relatives talk badly about them and mock them behind their backs and sometimes, even in their presence. One participant shared her experience:

*Last Christmas 25th December was my wedding anniversary and I wanted to have a little get-together. So, I discussed it with some aunties and family members and one person responded; what are you celebrating in your marriage? The person said that to remind me of my inability to give birth after marriage.* (Couple 005; 29-year-old; female partner with tertiary education; 5 years of infertility)

The study participants also recounted how anytime they get into an altercation with other family members, they are quickly reminded of their infertility. Their inability to conceive was often regarded as a "disability". In some cases, direct references were made to their infertility. This situation puts enormous psychological stress on infertile couples as they attempt to find solutions to their infertility. Of even greater concern to infertile couples was how some family members make snide remarks at family gatherings and whenever they visit the couple. A participant recounted:

*Last year July, I could not attend a send-off ceremony of my auntie's husband, and the following weekend we met at our family house. . . I tried to explain to her that it was because of work-related time constraints. But she was still arguing and she told me that, I should be the one who has more time than everybody because I don't have kids to take care of or worry about. I was completely broken* [participant started crying]. (Couple 009; with 9 years of infertility).

**Self-stigma.** Whereas the narratives point to the family being the most common source of psychological stress, we also found "self" as a source of stress for infertile couples. Most

infertile couples, especially female partners occasionally blamed themselves for their infertility. This was common among women with a history of miscarriages, those whose husbands already have children with other women, and those who have had tests revealing their infertility to be linked with past procedures such as unsafe abortion. In such cases, the source of stressors is the "self" (infertile women) as they actively question themselves and blame themselves for their predicament. This can be psychologically self-destructive for the infertile woman. A couple narrated their ordeal as:

> *Last week, we saw people with little children and we were sad [woman sighs]. It is not easy. So, at the workplace and social gatherings, when we come across people with children and pregnant women, we always ask ourselves why our case is different.* (Couple_004; 4 years of infertility)

Another participant had this to say: *Anytime I go for fertility consultation [clinic] and the doctors ask me about my past pregnancies and how many times I've been pregnant, and I have to tell the doctors that I did two abortions in the past, I feel like leaving the consulting room and never go again for fertility clinic because I am the cause of my current situation. If I had not done those abortions, I wouldn't have difficulty conceiving now.* (Couple 010; 33-year-old, female partner with tertiary education; 5 years of infertility)

**Stressors from society/community and work colleagues.** Another source of psychological stressors for infertile couples is their communities/society. Stressors from society often emanate from "jokes" made concerning infertility when they find themselves at social gatherings. Participants accentuated that often society assumes that married couples automatically have children because they have been married for a period. As such, people inadvertently ask about the whereabout or the health of the children when they come across married couples. There were, however, mixed expressions from the study participants regarding people's intentions. Whereas some participants regarded it as a form of stigmatization, others saw it to be just a mistake due to societal proscriptions of how married couples ought to be. However, study participants unanimously attested to the fact that their colleagues at their workplaces occasionally passed insensitive remarks to get them annoyed. The quote below attests to this theme:

> *I remember an incident which happened at work. My morning shift is normally supposed to end at 2 pm and the afternoon staff is supposed to take over immediately. Sometimes, there are a few minutes of delays which is understandable. One day a senior staff nurse who was supposed to take over delayed and came 30 minutes late. I told her it was wrong. Then she and another colleague replied to me saying that I don't have any children to pick from school or take care of, so I can't understand. Why am I making it a big deal. . .? Her reply got me so hurt emotionally.* (Couple 004; 27-year-old; Female partner with tertiary education; 4 years of infertility)

**Stressors from the church.** Of important note is that the church also emerged as a source of stressor for infertile couples. This is particularly concerning because there is documented evidence that points to the church as a place of comfort and preservation of life [18]. The study participants highlighted that Church sermons sometimes caused them extreme psychological stress, even occasionally pushing some of them to the brink. The narratives from the study participants point to some sermons from preachers, whether intentionally or unintentionally as a source of stress for infertile couples. A couple recounts their experience as:

*Any time there is a church program and there is preaching about the blessings from God, the pastors make it look like childlessness is a curse. I don't know whether they say those things intentionally or unintentionally.* (Couple 001; 35-year-old female partner with primary education; 8 years of infertility)

**Psychological impact of infertility.** The study participants reported being overwhelmed by their condition of being infertile and the attendant attempts/efforts to find solutions. Several of the participants described how their failure to find a solution to their infertility often left them perplexed, desperate, depressed and generally moody at work.

**Desperation.** Almost all the study participants shared the experience that their infertility has left them in a state of desperation. In the case of infertile women, this state of desperation increases as they age. Most of the women revealed that they have gone outside what they would have done ordinarily in their quest to achieve pregnancy. They are willing to do anything to get a child. A female participant had this to say:

*We visited most of the hospitals in [name of a city] for my infertility issues and every time we were told that we didn't have any problem, that we should go and have more sex. It has been like that for the past 6 years. I spent countless days in prayer camps and also tried several herbal treatments, with no success. I am even currently combining herbal preparations and orthodox drugs. I am ready to take anything and do anything to get pregnant. We are here today to start the process of artificial insemination, maybe it can work.* (Couple_006; 7 years of infertility)

**Depression.** Almost all the study participants accentuated that their condition caused them to have episodes where they were depressed. They sometimes lose their appetite due to constant anxiety over their condition. They had a predominantly sad mood and sometimes could not sleep at night. Participants also highlighted that they occasionally did not have the urge to engage in activities that were hitherto considered pleasurable to them. This is what one of them had to say:

*So, when I see someone has given birth, I go inside to cry. Sometimes, for like 2 days, I starve and cry and think of my plight.* (Couple 008; 35-year-old female with Secondary education; 5 years of infertility)

## Coping strategies adopted by infertile couples

**Self-motivation.** Self-motivation was found to be one of the common coping strategies of infertile couples, especially soon after diagnosis is confirmed. Most of the couples draw motivation from themselves. Most of them feel left alone and do not think others will understand their predicament. As such, they often keep to themselves and motivate themselves as a way of coping with their conditions. Most of the participants highlighted that sharing their predicaments with others will only compound their problems as they will not be able to offer any tangible solution. This is what a female with 5 years of infertility had to say:

*. . . I don't share my plight with anyone because the African setup does not value privacy. Even if a health worker is a family member, you can't trust them with your issues, because you may be the centre of discussion at the next funeral or family event. So, I keep things to myself and motivate myself.* (Couple_005, A 29-year-old female with tertiary education; 5 years of infertility)

**Active avoidance strategy.** As a coping strategy, some women narrated how they avoided attending social functions to prevent certain intrusive and uncomfortable questions about their childlessness. Social functions are often used as a place where people interact and get up close and personal with one another. As such, many of the participants highlighted that they avoid such functions to prevent people from getting up close and personal with them. This is what one of the study participants said:

*Last year, my hospital organised a Christmas tree party for the staff and their children, but I did not go. I have to pretend that I am not feeling well. I want to avoid such gatherings so that I don't have to answer unpleasant questions about my fertility.* (Couple_004; 4 years of infertility)

**United in grief: Support from each other.** Another coping strategy adopted by infertile couples is to unite in grief where they support each other. This strategy works better when the husbands are very supportive of their wives. The narratives showed that most husbands are understanding and willing to go for clinic appointments with their wives and are ready to undergo the necessary tests as part of the efforts to find solutions to the problem. A participant narrated:

*My husband has been very supportive. He encouraged me to come to the hospital and often comes to the hospital with me as you can see today. My husband considers the problem as our problem and has been very supportive. Anytime I am down, he supports me with words of encouragement.* (Couple_010; 5 years of infertility)

The narratives also point to infertile couples shielding each other from family and friends to avoid having to answer unpleasant questions about their predicament. Sometimes, they put up a posture to suggest that their infertility does not bother them. This is a deliberate coping mechanism to prevent people from prying on their issues. Participants highlighted that such coping mechanisms protect them from people who will constantly remind them of their infertility by suggesting solutions that would not even work. This is how one of the study participants put it:

*To avoid people always bringing up suggestions, we often create the impression that we are okay, and not bothered about it. We do this to protect ourselves from psychological stress. . . So, they will not try adding us to fertility pages [social media platforms] and finding solutions.* (Couple_006; 7 years of infertility)

**Active engagement in church and religious activities.** Some infertile couples admitted to seeking help from the church, including groups within the church. Whereas some participants see the church as the first option, others see it as the last resort in their attempts to solve their infertility problem. According to them, the church is the only place they can take their problems to receive both psychological and spiritual help. A couple recounted:

*During my uterine fibroid surgery 4 years ago, I had a lot of complications, I nearly died and my church members and pastors had to pray for me at the hospital and fortunately, God saved my life. So, I know he will grant me the fruit of the womb; my God is a prayer answering God. . . Since then, I have been more involved in church activities, mostly the miracle prayer meetings.* (Couple 002; 37-year-old; 9 years of infertility)

## Discussions

The current study seeks to explore the psychosocial burden of infertility by drawing from the perspectives of infertile couples in the Central region of Ghana. To the best of our knowledge, this is one of the few studies which focuses on the experiences of couples living with infertility in Ghana. Our study revealed that the psychological burden of infertility largely emanates from the family, the society, and self-inflicted, with the family being the main source of psychological stress to infertile couples. This finding is consistent with documented evidence from Iran that couples dealing with infertility face pressures from family, friends, their partners, as well as themselves [3].

Of note in the present study is the evidence that the family either deliberately or unintentionally exert enormous psychological stress on infertile couples. It was evident that close family relations often knowingly or unknowingly cause psychological stress for infertile couples. A study by Lei, et al., [19] reported that family adaptability has a negative correlation with infertility-related stress in women. Indeed, psychological stress as a result of infertility has been found to have adverse effects on a couple's quality of life and their relationship with the family [20]. Studies have also reported that women seem to experience more infertility-related psychological stress than their male partners [21, 22].

We also found that infertile couples often self-stigmatized which leads to psychological stress. Taebi et al., [3] reported that infertile women are faced with social and self-stigma which often negatively impact their psycho-social wellbeing and self-esteem. It is a predictor of anxiety, depression, and psychological distress [8]. This is consistent with this current study which also shows that most of the infertile women looked down on themselves because their fellow women have children while they are unable to achieve pregnancy. Taebi, et al., [3] reported that infertile women had feelings of incompleteness and defect.

Our study also revealed that society somewhat places stress on infertile women. According to a study conducted in Iran by Bakhtiyar, et al., [23] infertility does not have much social health impact on infertile women. The study reported that infertility has a psychologically and physically negative impact, but no considerable social impact. This is not consistent with what we found in the present study where infertility has considerable negative social impact, especially on women. The disparity could be due to differences in context as the Ghanaian culture and understanding of infertility may be different from that of Iran. The fact that infertile women experience more psychological stress than their male partners may be a result of gender roles and sex-role identification [24].

Traditionally, motherhood (the ability to produce live birth) is more closely linked to femininity than fatherhood is linked to masculinity [22]. A previous study conducted in Ghana also revealed that infertility has social implications for the affected couples. The social effects of infertility on couples include exclusion, verbal and physical abuse and divorce [25]. This is partly consistent with the findings of the current study where it was evident that infertile couples felt socially excluded, especially when it comes to the attitudes of their colleagues at the workplace. However, our study did not find any form of physical abuse, nor did we record any divorce, although, one out of the twenty participants feared her husband would leave her due to her infertility.

Also of note in the present study is the evidence that the church is a source of stressors for infertile couples. This finding is rather surprising given that Roudsari and Allan [26] reported that for infertile couples, religion provides an avenue to maintain hope and give meaning to their sufferings. Religious infertile women regard their condition/predicament as an enriching experience for attaining spiritual growth [26]. This disparity may be due to the previous study's emphasis on counselling as a key part of support for infertile women. Therefore, conscious

efforts (including pastors/preachers being sensitive to social issues) need to be made by the church if it wants to be a source of hope for infertile women.

Our study also revealed that infertility exerts an enormous psychological impact on couples which often results in desperation and depression, especially among infertile women. This is consistent with existing literature which highlights that infertility is associated with a myriad of psychological problems, including feelings of guilt, anxiety, and depression [27]. This highlights the need for psychotherapy even before medical treatment of infertility is initiated as studies have linked the failure of fertility treatment to psychological distress [28]. It is important to note that in the current study, the participants were infertile couples who were actively seeking treatment. As such, they were more likely to feel the psychological impact of infertility, leading to desperation and depression. The impact may not be so pronounced on couples who have resigned to fate and not actively seeking treatment for their infertility issues. Furthermore, in instances where the barrier to treatment is financial, couples that cannot afford the costs of treatment could have a more severe negative psychological impact than those able to pay for the treatment [27].

Several coping mechanisms were adopted by infertile couples in our study setting. This includes; self-motivation, active avoidance, united in grief by supporting each other, and active engagement in church and religious activities. Our study resonates with Taebi, et al., [3] who revealed that some women internalize the stigma and it takes a toll on their self-esteem and may lead them to social isolation. They also added that infertile couples have feelings of shame, inferiority, worthlessness and loss of control. This is because, in numerous settings, childbearing is seen as a symbol of virtue and prestige. Evidence from Turkey points to infertile women resorting to active involvement in religious activities as a coping strategy as they experienced stigma and hopelessness due to infertility [29].

The present study also revealed that as part of the coping strategy, infertile couples united in grief and provided support for each other. The evidence suggests that this strategy works better when husbands are supportive of their wives. This reinforces the misconception that women need support from men but not the other way around when the cause of infertility lies with the man. This misconception often heightened the psychosocial burden of infertility on women more than men [30].

It is important that infertile couples adopt appropriate coping strategies to deal with their infertility. Maladaptive coping strategies may not positively affect psychological health among infertile couples [31]. It is also important to note that women's self-blame and self-focused rumination strategies have a negative impact on women's psychological health [32]. In a study by Pedro, it was reported that infertile women had difficulties with friends with children and often experienced social withdrawal at social functions such as family and friends' gatherings due to feelings of being marginalized [33]. There is also documented evidence to suggest that infertile women deliberately avoided discussions related to family or children as a way of protecting themselves from uncomfortable conversations that could negatively impact their psychological health [34].

Theoretically, although the source of stigma seems to be mainly from the interpersonal level (i.e., family, friends and society), this study also showed that the same interpersonal-level relationship (especially the family) played an important role in helping infertile couples to cope with their situation. It is important to note that in instances where stress and tension appeared to be from the family, they were often not direct and usually unintentional. Again, while the church was a source of stress for infertile couples, it also served as an avenue for infertile couples to cope with their situation.

Even though our study offers important insight into the psychosocial burden of infertility, drawing from couples' perspectives, there are some inherent limitations. Our study

participants were recruited from a single fertility clinic in southern Ghana. Even though the study setting is multi-ethnic, we were not able to explore in detail the dynamics of the various cultural systems and their influence on the psychological health of infertile couples. It is also important to note that the experiences shared in this paper are exclusively from infertile couples attending a fertility clinic. Infertile couples who are not actively seeking solutions could have different experiences and perspectives on this important social phenomenon. However, we could not include them in this study as access to them is practically almost impossible due to social and cultural sensitivities surrounding infertility. Also, due to the qualitative design that was employed with a small sample size, the results cannot be generalized. However, these limitations do not diminish the importance of the evidence presented since our goal is to unravel the phenomenon rather than to generalise it.

## Conclusion

Infertility exerts a considerable negative psychosocial burden on couples, especially women. The interpersonal level which includes family, friends and colleagues serves as the main source of psychological stressors for infertile couples. Couples with infertility resort to reduced social interaction through the adoption of active avoidance as a strategy to cope with their problems. Infertility is not just a medical condition; it is also a deep-seated personal and socially embedded phenomenon. Therefore, programmes aimed at addressing fertility treatment should consciously consider addressing the psychosocial burden of infertility through education targeting actors, especially interpersonal-level actors. Also, fertility healthcare professionals should provide comprehensive care (including psychological therapy) to affected couples.

## Acknowledgments

The authors express appreciation to the management of the study health facilities and the study participants who shared their perspectives with us.

## Author Contributions

**Conceptualization:** Abdoul Azize Diallo, Prince Justin Anku.

**Data curation:** Rhodalyn Adwoa Darkoa Oduro.

**Formal analysis:** Abdoul Azize Diallo, Prince Justin Anku, Rhodalyn Adwoa Darkoa Oduro.

**Methodology:** Abdoul Azize Diallo, Prince Justin Anku, Rhodalyn Adwoa Darkoa Oduro.

**Writing – original draft:** Abdoul Azize Diallo, Prince Justin Anku.

**Writing – review & editing:** Abdoul Azize Diallo, Prince Justin Anku, Rhodalyn Adwoa Darkoa Oduro.

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
