## [Decision Letter · Decision Letter 0]

11 Oct 2023

PONE-D-23-28365Exploring the psycho-social burden of infertility: Perspectives of infertile couples in Cape Coast, GhanaPLOS ONE

Dear Dr. Abdoul Azize,

Thank you for submitting your manuscript to PLOS ONE. After careful consideration, we feel that it has merit but does not fully meet PLOS ONE’s publication criteria as it currently stands. Therefore, we invite you to submit a revised version of the manuscript that addresses the points raised during the review process. The reviewers have recommended that major revisions be made to your paper in order to meet the publication criteria. As indicated by both reviewers, the methods and data analysis sections require significant revision to be clearer and more critical to show how the findings were reached. There must also be a critical discussion on the implications of the findings for couples facing infertility and a reflection on the wider utility of suggested interventions for other couples. Please ensure that this important distinction is made throughout the paper. A thorough proofreading of the manuscript is also necessary. Please submit your revised manuscript by Nov 25 2023 11:59PM. If you will need more time than this to complete your revisions, please reply to this message or contact the journal office at plosone@plos.org. Please include the following items when submitting your revised manuscript:A rebuttal letter that responds to each point raised by the academic editor and reviewer(s). You should upload this letter as a separate file labeled 'Response to Reviewers'.A marked-up copy of your manuscript that highlights changes made to the original version. You should upload this as a separate file labeled 'Revised Manuscript with Track Changes'.An unmarked version of your revised paper without tracked changes. You should upload this as a separate file labeled 'Manuscript'.

We look forward to receiving your revised manuscript.

Kind regards,

Lily Kpobi, Ph.D.

Academic Editor

PLOS ONE

We will update your Data Availability statement to reflect the information you provide in your cover letter."

Reviewers' comments:

Reviewer's Responses to Questions

**Comments to the Author**

1. Is the manuscript technically sound, and do the data support the conclusions?

Reviewer #1: Yes

Reviewer #2: Yes

2. Has the statistical analysis been performed appropriately and rigorously? 

Reviewer #1: N/A

Reviewer #2: N/A

3. Have the authors made all data underlying the findings in their manuscript fully available?

Reviewer #1: Yes

Reviewer #2: Yes

4. Is the manuscript presented in an intelligible fashion and written in standard English?

Reviewer #1: Yes

Reviewer #2: Yes

5. Review Comments to the Author

Reviewer #1: The authors presented the psycho-social impact of infertility in affected couples in a qualitative study, using a phenomenology approach. Data was collected from 10 couples attending fertility clinic in Feb 23 in Cape Coast, Ghana.

The paper is generally well written, the structure is good and clear and the topic is relevant to health care providers in fertility treatment as psycho-social burden on the affected couples should be addressed concurrently with the actual treatment. Therefore, results from this study can inform health care professionals to provide better care.

The population is small, and the results are not applicable to the general infertile population due to various reasons (some of which addressed in the discussions), including the fact these couples were specifically seeking fertility treatment. However, this study population is certainly an important subset more likely to benefit from possible interventions, compared to couples not reaching out for help and support, due to the complexity and sensitivity of the problem.

Further comments:

1- In the abstract where it says, “we explored the impact of infertility on the psychological and social health of infertile couples” I would add “receiving fertility treatment”. Similarly, in other relevant places of the manuscript, it should be highlighted that the targeted population was not general infertility couples but the ones attending fertility clinics.

2 -The Social Ecological Model (SEM) - can the author explain better the difference between interpersonal, organizational and community please? as at the moment they seem to overlap.

3 - Can the authors add whether fertility treatment in Ghana is supported by the National health system or whether the affected couples need to pay for treatment as this would clarify better the socio-economic status of the interviewed couples.

4 - Can the author expand on further bias in the selected population, in the discussion section. For example, that these people were proactively addressing the infertile issue and were willing (and able) to seek help. Would this attitude come from a more or less ‘desperate’ population? And how easy/difficult would it be to recruit a more general infertile population? And what are the implications on the results of this study, by using this restricted population?

5 - Page 15 - “This strategy works better when the husbands of the various couples are very supportive of their wives”. This statement seems to highlight that women need support from men but not the other way around when the cause of infertility is with the man. Can the authors expand a bit on this in the discussion section? is it a preconception that women need a husbands’ support as they are considered the main cause of infertility but not the other way around?

6 - Previous literature has reported the difference in perception and attitude towards infertility between developing and developed countries. Can the authors discuss this and if the study confirmed any of these differences?

7 - In the conclusion, I think one important message, other than educating interpersonal-level actors, is also for the health care professionals to provide more comprehensive care (including psychological) to the affected couples.

8 - Table 2 - can the authors add the Frequency of sub-theme in an added column please?

9 - page 11- “self-distractive” did the authors mean “self-destructive”?

10 - Page 20 - “although the source of stigma and coping strategies” I suggest deleting the text “and coping strategies” to make sense of the phrase.

Reviewer #2: Manuscript Number PONE-D-23-28365

"Exploring the psycho-social burden of infertility: Perspectives of infertile couples in Cape Coast, Ghana"

- Aim of the study not clearly identified; this needs to be specified in the study background.

- Methods – it is not specified which phenomenological approach was used for the study – descriptive or interpretive with clear descriptions of its respective philosophical underpinnings and how that was used as guide for the data collection and analysis. The research design fits an exploratory-descriptive study

- Minor phrase correction Table 2: Themes and subthemes from the infertile couples; Subthemes – need to be numbered.

-In the data analysis, how were the themes and subthemes generated noting that a model was used for the research? How was the Social Ecological Model (SEM) used as the organizing framework for the research noting that study themes are largely predetermined when models/frameworks are used in research; it is unclear how the constructs of the existing model guided the development of the current themes/subthemes which are different from the constructs of the model.

- Explanatory/analytic statement have been provided for some quotes but not all; these transition statements need to be provided between participant quotes before moving on to other themes and subthemes and subsequent headings.

-Study limitations - Explain measures to address language translation implications on the findings of the study, its inherent dynamics, how it influences participants' narratives and the meanings of the text. It is important to note potentially lost meanings of the translated text since the interviews were conducted in the local language and translated into the English language. Within the Ghanaian local language, there are words or phrases or expressions that do not have its direct translations into English.

- Minor sentence correction - The authors express appreciation to the management of the study health facilities and the study participants shared who their perspectives with us.

6. PLOS authors have the option to publish the peer review history of their article (what does this mean?). If published, this will include your full peer review and any attached files.

Reviewer #1: No

Reviewer #2: No

---

## [Author Response · Author response to Decision Letter 0]

24 Oct 2023

Response to reviewers file has been uploaded.

---

## [Decision Letter · Decision Letter 1]

12 Dec 2023

PONE-D-23-28365R1Exploring the psycho-social burden of infertility: Perspectives of infertile couples in Cape Coast, GhanaPLOS ONE

Dear Dr. Abdoul Azize,

Thank you for submitting your manuscript to PLOS ONE. After the second round of reviews, Reviewer 1 has recommended important minor revisions before this manuscript can be taken forward. Therefore, we invite you to submit a revised version of the manuscript that addresses the points raised during the review process.

The reviewer raises an important point about the differences in adverse psychological reactions based on ability to afford treatment or otherwise. This is important to reflect on and discuss.

We look forward to receiving your revised manuscript.

Kind regards,

Lily Kpobi, Ph.D.

Academic Editor

PLOS ONE

Journal Requirements:

Reviewers' comments:

Reviewer's Responses to Questions

**Comments to the Author**

1. If the authors have adequately addressed your comments raised in a previous round of review and you feel that this manuscript is now acceptable for publication, you may indicate that here to bypass the “Comments to the Author” section, enter your conflict of interest statement in the “Confidential to Editor” section, and submit your "Accept" recommendation.

Reviewer #1: (No Response)

Reviewer #3: All comments have been addressed

2. Is the manuscript technically sound, and do the data support the conclusions?

Reviewer #1: Yes

Reviewer #3: Yes

3. Has the statistical analysis been performed appropriately and rigorously? 

Reviewer #1: N/A

Reviewer #3: Yes

4. Have the authors made all data underlying the findings in their manuscript fully available?

Reviewer #1: Yes

Reviewer #3: Yes

5. Is the manuscript presented in an intelligible fashion and written in standard English?

Reviewer #1: Yes

Reviewer #3: Yes

6. Review Comments to the Author

Reviewer #1: My only comment is on the edited text on page 19, paragraph 2, about participants being infertile couples who were actively seeking treatment. I think that couples who have resigned to fate might have less adverse psychological impact than the ones seeking treatment, however the blocker to treatment access could also be financial. Couples that cannot afford to pay for the treatment could have a more severe psychological impact than the ones able to pay for it. I suggest the authors add this potential option too.

There is typo on page 20, 2nd paragraph, edited text - “cooing strategy” should be “coping strategy”.

The authors addressed all my other comments in a satisfactory manner.

Reviewer #3: Thank you for the opportunity to review this paper on the important subject of infertility in Ghana.

Thank you for your responses to the comments made by the reviewers.

Congratulations!

7. PLOS authors have the option to publish the peer review history of their article (what does this mean?). If published, this will include your full peer review and any attached files.

Reviewer #1: No

Reviewer #3: **Yes: **Timothy Tienbia Laari

---

## [Author Response · Author response to Decision Letter 1]

15 Dec 2023

This has been uploaded as a separate file

---

## [Editor Report · Decision Letter 2]

5 Jan 2024

Exploring the psycho-social burden of infertility: Perspectives of infertile couples in Cape Coast, Ghana

PONE-D-23-28365R2

Dear Dr. Abdoul Azize,

We’re pleased to inform you that your manuscript has been judged scientifically suitable for publication and will be formally accepted for publication once it meets all outstanding technical requirements.

Kind regards,

Lily Kpobi, Ph.D.

Academic Editor

PLOS ONE

---

## [Editor Report · Acceptance letter]

17 Jan 2024

PONE-D-23-28365R2 

PLOS ONE

Dear Dr. Azize Diallo, 

I'm pleased to inform you that your manuscript has been deemed suitable for publication in PLOS ONE. Congratulations! Your manuscript is now being handed over to our production team.

Kind regards, 

on behalf of

Dr. Lily Kpobi 

Academic Editor

PLOS ONE